# Personal Protective Equipment Portraits Canada (PPC)–Humanization and surveying mask-wearing nationally

**Laura Rendon**[1], **Tarek Taifour**[1], **Cynthia R. Ventrella**[1], **Ana Seara**[2], **Adamo A. Donovan**[1]*

**1** Faculty of Medicine and Health Sciences, McGill University, Montréal, Canada, **2** Veterans Centre, Sunnybrook Health Sciences Centre, Toronto, Canada

* adamo.donovan@mail.mcgill.ca

**Data Availability Statement:** All relevant data are within the paper and its Supporting Information files.

## Abstract

### Background

Personal Protective Equipment (PPE) Portraits is a hybridized art and medical intervention that lessens the alienating appearance of PPE through wearable, smiling headshot pictures. During the pandemic, the use of these portraits was expanded, but Canadian initiatives offered portraits only to immediate stakeholders. PPE Portraits Canada (PPC) aimed to provide PPE portraits to any Canadian healthcare institution and surveyed healthcare workers (HCW) regarding these portraits' impact.

### Methods

University student volunteers founded PPC via online platforms and coast-to-coast collaborations that allowed any HCW nationwide to request a free portrait via an accessible online form. PPC has gathered feedback from participating HCWs directly via an anonymous and bilingual survey.

### Results

70% of HCWs wore their portraits "*always*" or "*usually*", 69% of HCWs "*definitely would*" recommend their portrait, 89.5% of HCWs found that the PPE portraits made a difference in their experiences with patients and 74% found the same for their colleagues. The pre- and post-effect of the portraits, led to a 37.5% greater likelihood that HCWs felt "*connected*" or "*very connected*" to patients/residents. For the thematic analysis, 70% or more of the comments were rated as positive, with less than 5% of comments being rated as negative.

### Conclusion

This model's logistical framework can be expanded beyond PPE portraits to other initiatives with limited resources, allowing them to reach and positively impact diverse populations. HCW feedback was predominantly positive. The optimal design and impact of PPE portraits on patients and HCWs should be studied further to improve portrait adoption.

**Funding:** The authors received a Practiced-based Research and Innovation (PBRI) Special Seed Grant from Sunnybrook Health Sciences Centre to cover the costs of the survey implementation and Survey Monkey subscription. This funder approved the original research proposal, and thereafter was in a supportive role. To pay for all fees associated with PPE Portraits Canada's informational technology backend and the logistics supporting the request, purchase, and delivery of portraits, the authors received the following: - 2020 McGill Mary H Brown Fund - 2020 TakingITGlobal #RisingYouth grant - 2020 Surrey Memorial Hospital Engagement Project Fund - 2021 University of Toronto Pillar Sponsorship Program - 2021 Research Institute of the McGill University Health Centre (MUHC) Desjardins Centre for Advanced Training (DCAT) Trainee-led Initiative Funding - 2021 Ontario Medical Student Association (OMSA) Innovator Grant These funders had no role in study design, data collection and analysis, decision to publish, or preparation of the manuscript.

**Competing interests:** This paper mentions several softwares and programs utilized by the authors and the PPE Portraits Canada organizing team. Both parties have benefitted from non-profit or COVID-19 response discounts and/or free premium subscriptions to use these programs at scale. However, these are non-exclusive and none of the authors have received any monetary compensation nor are they affiliated in any other way with the companies behind these products. Exceptionally, PPC received an exclusive sponsorship from a local printing sponsor in Montréal (The Business Box) that is mentioned in the article to receive PPE Portraits at cost. The authors are in no way obligated to mention these programs but are doing so to inspire others and facilitate the scaling of other initiatives. This does not alter our adherence to PLOS ONE policies on sharing data and materials.

## Introduction

The introduction of necessary Personal Protective Equipment (PPE) guidelines to combat the COVID-19 pandemic in March 2020 served as an additional source of stress for patients, their loved ones, and healthcare workers (HCWs). This equipment masked facial expressions and made human interactions alienating, fostering patient fear, mistrust, and negatively impacting the recovery and overall health of patients [1]. In April 2020, inspired and guided by Mary Beth Heffernan's PPE portrait work during the 2014–16 African Ebola epidemic and early COVID response, a group of Montréal students began offering smiling headshot portrait badges to HCWs that they can fix to their person and/or over their PPE [2]. Despite limited research, PPE portraits have received positive feedback from the majority of HCWs and patients. They have been shown to (1) improve HCW's mood and sense of well-being, (2) help with team dynamics, aide colleague identification, enhance perceptions of team connection, and increase interactions with fellow HCWs, (3) improve patient interactions through increased comfort of frontline HCWs when interacting with patients and (4) increase patient happiness when interacting with HCWs in PPE [3].

Due to restrictions and the early uncertainties of the pandemic, this group respected social distancing protocols and collected portraits via self-submitted pictures received through an online form, while limiting the financial and time burdens that would be imposed on the strained availabilities and resources of frontline workers. Automation of the system allowed to minimize both HCWs' and the organizing team's workloads and increase efficiency. With early successes in Montréal and efficient workflows, the team made portraits available country-wide through collaborations with other students who understood how to overcome their local promotional barriers. Thus, PPE Portraits Canada (PPC) was formed [4]. PPC would collaborate with Sunnybrook Health Sciences Centre to survey these HCWs and assess the impact of these portraits on HCWs, team dynamics, and patients, as well as assess potential obstacles and areas of improvement through both a quantitative and thematic analyses.

## Methods

Smiling, headshot portraits were submitted by HCWs and/or their institutions at their convenience to the PPC team via a bilingual (English and French) online form. For large institutional orders, organizations could email PPC the appropriate information and picture files to avoid having to fill out the form multiple times for each HCW. This form allowed for a remote and scalable manner for student volunteers from across Canada to collaborate, implement, and grow this project nationwide. The innovation, which provided country-wide accessibility, was achieved through a workflow (Fig 1 and S1 Video) that utilizes several programs to facilitate and increase the efficiency of PPC's logistical setup. These programs are easy to use and present low risk with high rewards, given that most are flexible, non-coding, and/or are free to minimal cost, especially when factoring in non-profit/educational discounts [5]. This contributes to increased professionalism, organization, and timesaving through automations that eliminate mistakes and that increase stakeholder appreciation and promotion of PPC through word of mouth [5]. To the best of our knowledge, PPC is the largest provider of PPE portraits, with over 3700 portrait requests from 27+ different professions in 26 Canadian cities across 5 provinces.

In addition to the portraits themselves, PPC has been involved in art installations in healthcare facilities that commemorate all frontline workers and those who contributed to the positive impact of the project during the pandemic. In collaboration with the Arts and Heritage Center of the McGill University Health Center (MUHC), the PPC team created mosaics displayed across multiple MUHC sites and locations, featuring the smiling submitted pictures of

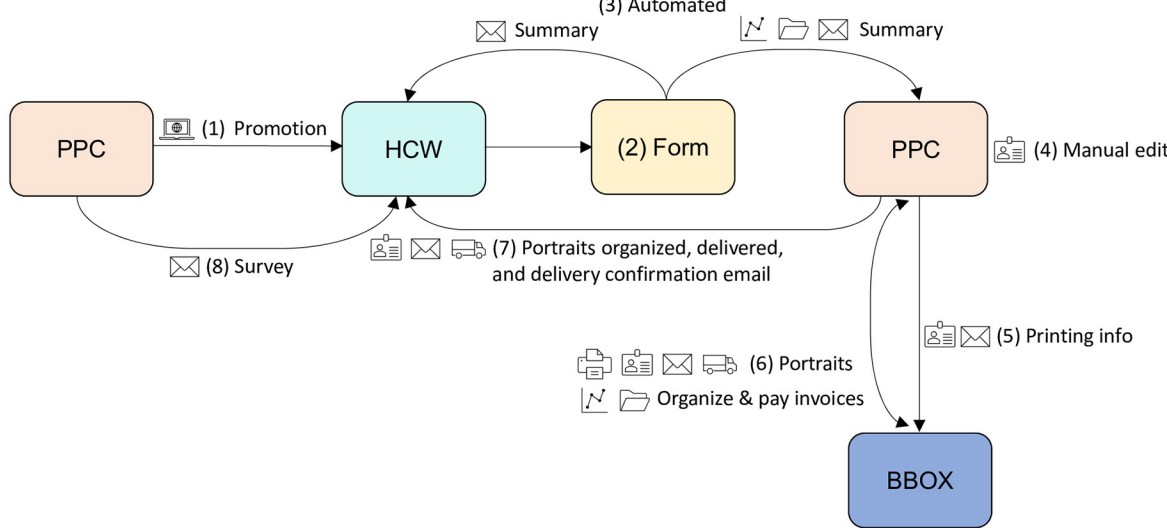

**Fig 1. PPE Portraits Canada workflow.** The workflow described here utilizes several programs that facilitate and increase the efficiency of PPC's logistical setup that provides portraits Canada-wide and the collection of survey data. This maximizes both the student volunteers and frontline workers' time and resources [5]:

1) Promotion: The PPC team actively promotes PPE portraits using a bilingual online presence which includes social media, search engine optimization (SEO), and a bilingual and dual desktop/mobile compatible website. Due to the powerful visual aspects of the portraits, one of the most effective tools for promotion was word of mouth and real-world demonstration when HCWs see their colleagues sporting their own portraits. To amplify this further, we facilitated form sharing by creating a simple and short branded URL and associated QR code using Rebrandly, that were printed on the portraits' backs, along with the supporters' logos (Fig 6).

2) Submission form: HCWs submit basic information in either English or French and a smiling picture of themselves using JotForm, whose conditional logic allowed us to cover specific and detailed requests from any of Canada's 10 provinces and 3 territories with the use of only 4 forms.

3) Automation: Once submitted, JotForm automatically sends an email to the HCW and PPC with a summary of the HCW's submission. The information is automatically uploaded to an Airtable database. The picture is organized in an automatically renamed folder (according to the HCW's name and location) and stored in a specific location in PPC's Google drive for file storage.

4) Manual edit: The PPC team resizes the pictures using Canva (Fig 6). The editing is done for two reasons:

1) For the printing sponsors' size specifications.

2) To emphasize the facial features and focus the attention on the warm and/or smiling expression, rather than any external and distracting backgrounds that may be present [2].

5) <u>Printing info:</u> Portraits and printing information are emailed to PPC's printing partner, The Business Box, a fast and organized local business based in Montréal that offered at-cost laminated portraits and were able to handle orders that differed in portrait size and backings based on location.

1) <u>Backings:</u> These reflected different local and/or national supporters, had a specific QR code and a memorable, branded, and shareable URL that directed HCWs to the appropriate portrait request form to facilitate word of mouth referrals (Fig 6).

2) <u>Size:</u> Due to local preferences, portraits differed in size and were either 2.5 inches wide by 4 inches tall (2.5"x4", smaller format) or 3 inches wide by 5 inches tall (3"x5", larger format) (Table 1). For durability purposes, the portraits were 7mm thick and, from informal feedback gathered, lasted approximately 1 year with heavy use and cleaning.

6) <u>Portraits:</u> The Business Box prints the portrait and informs the PPC team via email for pickup or direct shipment. The PPC team pays the invoices and updates the financial records in Airtable.

7) <u>Portraits organized and delivered:</u> The portraits and badge clips (for fixing the portrait to their person) are organized and prepared for final hand or postal service delivery. In the latter case, sender and recipient addresses are printed directly onto envelopes efficiently using Microsoft Word's native mailing feature. The estimated cost to get a portrait in the hands of a HCW was $3.44, including the lamination (offered at-cost by The Business Box), clip, envelope, stamp, and taxes. Due to the clip, the envelope was categorized as non-standard or oversize mail and was the most expensive element accounting for 67% of the total post-tax cost. Institutional bulk orders, where portraits and clips could be packaged and sent together to one location, would eliminate the need for the stamps and significantly reduce the cost per portrait, but would require a responsible individual(s) at the receiving end to distribute these upon arrival. Once delivered or sent out, mail merge is used to send a personalized, bilingual, standardized email to the HCWs informing them that the portrait has been delivered or is on its way.

8) <u>Survey:</u> At a minimum of 1 month after having sent out the portraits, mail merge is used to send a personalized, bilingual, standardized email to the HCWs providing a link to the anonymous and bilingual Survey Monkey research survey.

consented HCWs as pixels forming the image of a masked worker wearing their own portrait (Fig 2) [6]. This consent was obtained when HCWs requested a portrait via our online form; we explained how we would potentially use their image that was devoid of any identifying information prior to presenting the yes/no question: "*Do you consent to having your picture being used in a PPE Portraits Canada art installation at an exhibit*?".

## Research design/participants/procedures/validity/credibility

In collaboration with Sunnybrook, a 16-question bilingual anonymous pilot survey was sent to PPE portrait recipients at a minimum of 1 month after having mailed the portraits. This survey was developed from previous research that collected data from a COVID-19 testing site [7]. AS created the first draft, received feedback from her interdisciplinary team at the Sunnybrook Health Sciences Centre, and further edits were given by AAD. This evaluation was performed as a quality improvement initiative and therefore institutional ethics board review was not required as determined by the Sunnybrook Research Ethics Office. Consent was implied via completion of the optional online survey, as identifiable information was not collected at any point. The individuals in Figs 2 and 6 in this manuscript have given written informed consent (as outlined in PLOS consent form) to publish these case details.

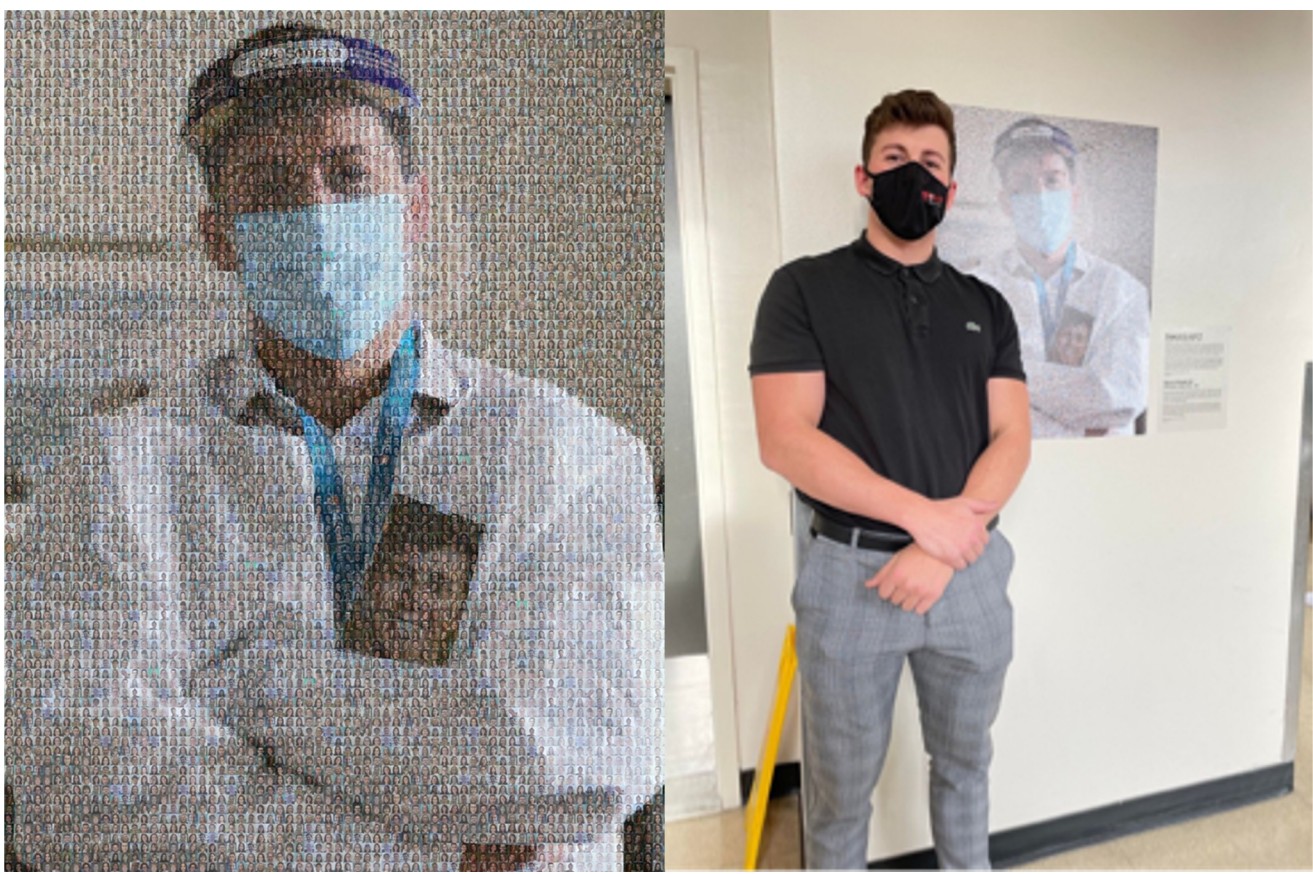

**Fig 2. PPE Portrait mosaic art installation.** This mosaic was created in collaboration with the Art and Heritage Centre of the McGill University Health Centre (MUHC) to recognize, thank and commemorate all frontline workers and to those who contributed to the positive impact of PPE Portraits Canada. The PPE headshot portraits of MUHC workers are featured here as pixels coming together to form the image of a healthcare professional at the Montréal General Hospital. This is one of 3 mosaics installed within MUHC hospitals (picture used with permission from the healthcare worker).

### Patient and public involvement

Patients were not involved in this study.

**Measures.** This survey consisted of 3 demographic questions (province, institution, and profession) for the descriptive analysis (Table 1), 8 multiple-choice questions for the quantitative analysis Figs 3–5 open-ended commentary questions for the thematic analysis (Fig 5 and S1 Table).

### Data analysis

For the thematic qualitative analysis, each response was first assigned 1 response type (positive, negative, or indifferent), 1 theme, and a maximum of 2 subthemes (Fig 5 and S1 Table). This was initially generated and assigned by LR before TT and CRV confirmed the assigned response types, themes, and subthemes.

### Results

Out of 3707 requests, 640 HCWs were not sent the survey, due to (1) not having received a portrait (missing or incorrect address, non-responsive to follow-up emails, recent portrait request, etc), (2) the absence of email information, (3) incorrect email (bounced), or (4) an

**Table 1. Portrait formats and requests, survey responses, and the language and professions of respondents organized by province/city.**

| Province City Portrait sizing | Atlantic Canada Manitoba Alberta 3.5"x5" | Quebec 3.5"x5" | Ontario | | | | British Columbia 2.5"x4.25" | Total = 3707 (100%) 3.5"x5" = 1159 (31%) 2.5"x4.25" = 2548 (69%) |
|---|---|---|---|---|---|---|---|---|
| | | | Ottawa 3.5"x5" | Hamilton 3.5"x5" | London 2.5"x4.25" | Toronto/ Other 2.5"x4.25" | | |
| **Portrait requests- N (%)** | 13 (0.5%) | 951 (25.5%) | 119 (3%) | 76 (2%) | 35 (1%) | 1967 (53%) | 546 (15%) | |
| **Reasons for not receiving survey (N)** **No Email** **Survey email blocked/bounced** **Portraits not delivered** | Total = 1 0 0 1 | Total = 269 203 20 46 | Total = 1 0 0 1 | Total = 3 2 0 1 | Total = 1 0 0 1 | Total = 326 277 42 7 | Total = 39 0 7 32 | Total = 640 482 69 89 |
| **Total potential respondents- N (%)** | 12/0.5% | 682 (22%) | 118 (4%) | 73 (2.5%) | 34 (1%) | 1641 (53.5%) | 507 (16.5%) | Total = 3067 (100%) 3.5"x5" = 885 (29%) 2.5"x4.25" = 2182 (71%) |
| **Survey respondents- N (%)** | 2/1% | 41 (22%) | 124 (67%) | | | | 19 (10%) | Total = 186 (100%) 3.5"x5" = 57 (31%) 2.5"x4.25" = 129 (69%) English = 175 (94.09%) French = 11 (5.91%) Response rate = 6.06% |
| **Survey respondents by profession- N (%)** | Trainee = 43 (23.11%) Therapist = 39 (20.97% Nurse = 37 (19.89%) Doctor = 29 (15.59%) Volunteer = 14 (7.53%) Other = 14 (7.53%) Administration = 6 (3.23%) Research = 4 (2.15%) | | | | | | | |

institutional email server that blocked external emails (Table 1). For those that received an email, it is unknown whether institutional servers blocked and rendered the Survey Monkey link inaccessible to prevent security breaches. The recruitment period was April 1, 2021, to December 31, 2022. Of the 3067 potential respondents, 186 individuals answered the survey, thus a 6.06% response rate (Table 1), and spent a median time of 3 minutes and 9 seconds filling out the survey.

Summary of the proportion of portrait requests, potential survey respondents, and survey respondents by province and portrait size:

| | $\frac{(Provincial)\ Portrait\ requests}{(Total)\ Portrait\ requests}$ | $\frac{(Provincial)\ Potential\ survey\ respondents}{(Total)\ Potential\ survey\ respondents}$ | $\frac{(Provincial)\ Survey\ respondents}{(Total)\ Survey\ respondents}$ |
|---|---|---|---|
| Atlantic Canada/Manitoba/Alberta | 0.5% | 0.5% | 1% |
| Quebec | 25.5% | 22% | 22% |
| Ontario | 59% | 61% | 67% |
| British Columbia | 15% | 16.5% | 10% |
| 3.5"x5" | 31% | 29% | 31% |
| 2.5" x 4.25" | 69% | 71% | 69% |

*Potential survey respondent = (Portrait requests) − (Healthcare workers who did not receive the survey)*

Due to small samples, professions were grouped into categories:

- <u>Administrative:</u> Inpatient or ward clerk, agent, clinical program assistant, executive assistant, or manager.

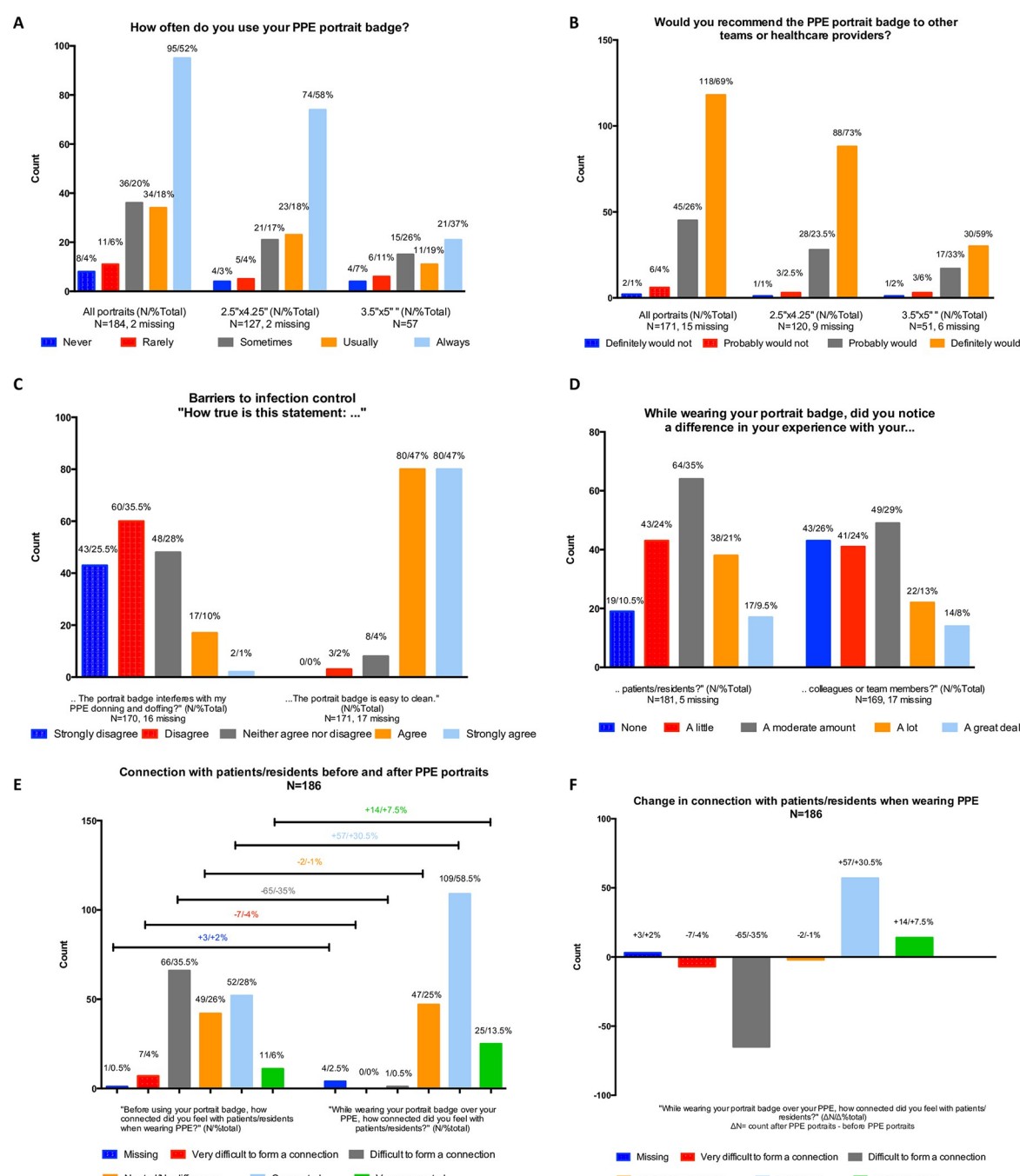

**Fig 3. Survey results summary measuring experiences of health care workers with portrait badges by badge size.**

- Doctor: Dentist or physician (obstetrics and gynecology, cardiology, otolaryngology, developmental pediatrics, family medicine, or psychiatry).

- Other: "*Anesthesia*", cook/chef, housekeeping, medical radiation technologist, physician assistant, midwife, patient care assistant, personal support worker, pharmacist, orderly, professor, respiratory therapist, or technician.

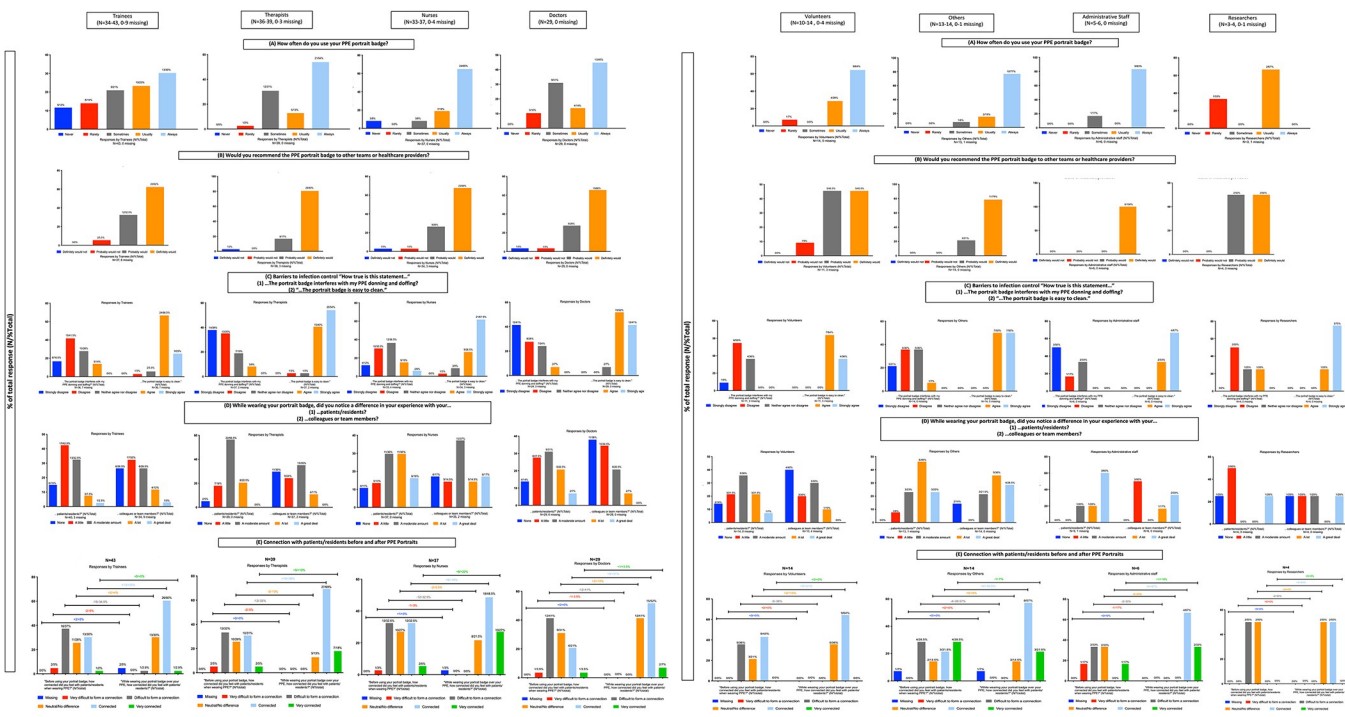

**Fig 4. Survey results summary measuring experiences of health care workers with portrait badges by profession.**

- Nurse: Registered nurse, lactation consultant registered nurse, public health nurse, registered psychiatric nurse, nurse educator, assistant nurse manager, nurse clinician, clinical nurse specialist, nurse practitioner, or clinical extern.

- Research: "*Research*", Clinical research study coordinator, research associate, or graduate student.

- Therapist: Activity aide, behaviour therapist, art therapist, music therapist, occupational therapist, physiotherapist, recreational therapist, rehabilitation therapy, social worker, speech language pathology, chaplain, mental health clinician, spiritual care provider, or special education instructor.

- Trainee: Fellows, residents, "*student*", or medical/dentistry/nursing/physiotherapy student.

- Volunteer: Volunteer or volunteer coordinator.

The survey sample was generally representative of the total number of portrait requests and the total number of potential respondents by both sizing and province/city. There was a slight overrepresentation of Ontario HCWs (6% relative increase to the potential respondents and 8% increase from the total number of portrait requests) and underrepresentation of HCWs from British Columbia (6.5% decrease relative to the potential respondents and 5% decrease from the number of portrait requests) (Table 1).

For the quantitative analysis, 70% of HCWs wore their portraits "*always*" or "*usually*", however this number increased to 76% when subsetting for HCWs that received a smaller format portrait and decreased to 56% with the larger format (Fig 3A). Furthermore, 69% of HCWs "*definitely would*" recommend a PPE Portrait to others, and similarly, this increased to 73% for the smaller portrait and decreased to 59% for the larger format (Fig 3B). In general, more than 90% of HCWs "*definitely would*" or "*probably would*" recommend the PPE Portraits

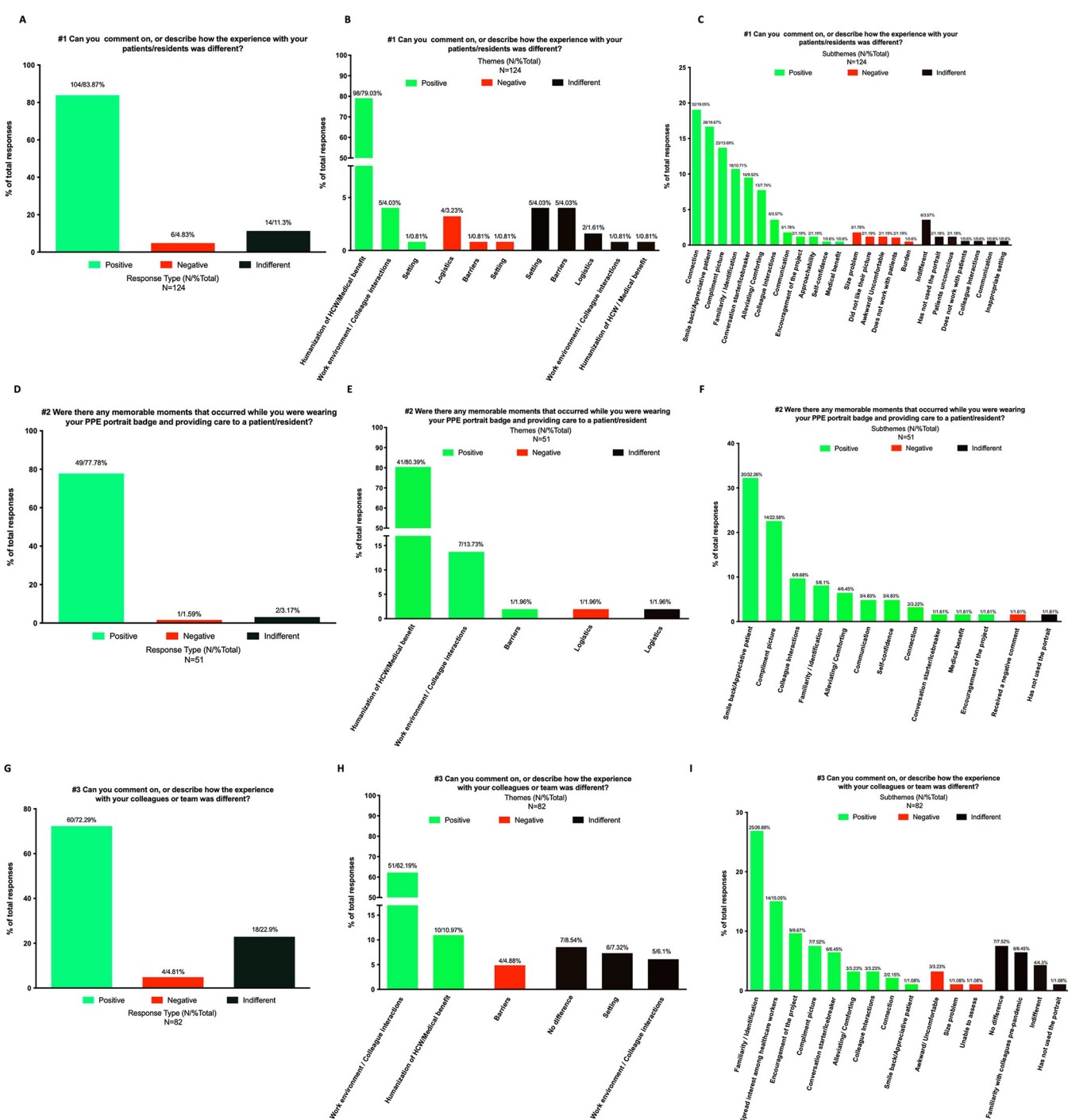

**Fig 5. Thematic analysis and meaningful examples.**

| Open-Ended Questions | Meaningful examples |
|---|---|
| **#1**: Can you comment on, or describe how the experience with your patients/residents was different? | "Nice to show the face that lives under the PPE. Humanize the experience patients are going through. I smile a lot and my picture allows me to share this with patients outside of the crinkle in my eyes when smiling with full PPE"<br>• Positive<br> ○ Humanization of HCW / Medical benefit<br> ■ Connection |
| | "I did not end up wearing the badge. It was printed larger than my ID and proximity access cards, and ended up hanging down lower than them and displaying the logo of a hospital/health authority that I don't work at, which I thought would create confusion. I would suggest printing it to the standard size of hospital ID cards."<br>• Negative<br> ○ Logistics<br> ■ Size Problem<br> ■ Familiarity/Identification<br>"I did not wear it because we were asked NOT to smile so the patients don't feel we are laughing at their situation but some people did smile and I liked their picture better than mine so I never wore mine. I do not go into patient's room so it did not have the impact it did with the bedside nurses."<br>• Negative<br> ○ Setting<br> ■ Does not work with patients |
| | "I didn't find that I would refer to my portrait photo often, but I have noticed patients looking at it while i work with them."<br>• Indifferent<br> ○ Barriers<br> ■ Indifferent |
| **#2**: Were there any memorable moments that occurred while you were wearing your PPE portrait badge and providing care to a patient/resident? Feel free to share a story, or a direct comment someone shared with you. | "I saw patients and families relaxing, calming down and I always felt that patients and families had a better sense of who they were being taken care by"<br>• Positive<br> ○ Humanization of HCW / Medical benefit<br> ■ Alleviating/ Comforting<br> ■ Familiarity / Identification<br>"A patient said to me that it was good to see what my face looked like and liked to see me smile. People appreciated knowing what I look like under my mask."<br>• Positive<br> ○ Humanization of HCW / Medical benefit<br> ■ Smile back / Appreciative patient<br> ■ Familiarity / Identification<br>"Dying COVID patient in the ICU isolation. No family member allowed inside but me the nurse holding his hand and most likely the last face he saw was my PPE portrait."<br>• Positive<br> ○ Humanization of HCW / Medical benefit<br> ■ Alleviating/ Comforting<br>"I really like this idea, however I feel somewhat shy to wear it since most people don't show their faces, even though I do want to make that connection and make patients feel comfortable. I will try to find a good spot on my outfit to have the picture"<br>• Positive<br> ○ Barriers |
| | Negative = None |
| | Indifferent = None |
| **#3**: Can you comment on, or describe how the experience with your colleagues or team was different? | "Other team members would recognize and remember me more readily. The PPE portrait would spark conversations among team members that I believe wouldn't have otherwise happened."<br>• Positive<br> ○ Work environment / Colleague interactions<br> ■ Familiarity / Identification<br> ■ Conversation starter/Icebreaker<br>"They would approach me more than before often asking about where I got the PPE badge. It was a great conversation starter."<br>• Positive<br> ○ Work environment / Colleague interactions<br> ■ Spread interest among HCW<br> ■ Conversation starter/Icebreaker<br>"Very few on my unit (NICU) were aware of the PPE Portrait project. Since I have mine, many people of come to me to ask how they can get one as well. I see them more and more now!"<br>• Positive<br> ○ Work environment / Colleague interactions<br> ■ Spread interest among HCW |
| | "Many colleagues have commented on what a great photo I have (thank you Ana!), and also how easy it is to see my name on the badge, and identify me. I've tried promoting the PPE badge service, alas, most colleagues express that they don't want to have such a large photo of themselves (I have the medium size), or that they are not photogenic. I have also unfortunately had team members ask why I had such a humongous photo of myself, to which I patiently explained that it was for our patients and families to be able to identify me and see the face of the physician they were talking to."<br>• Negative<br> ○ Barriers<br> ■ Size problem<br> ■ Awkward/Uncomfortable |
| | Indifferent = None |

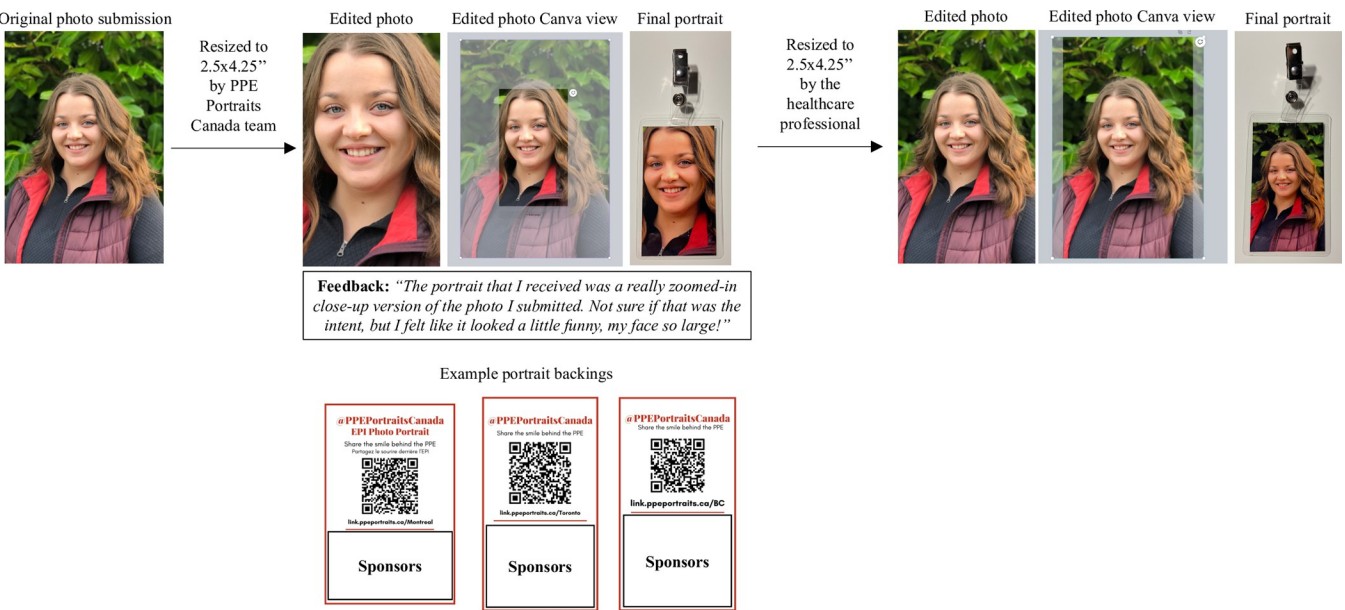

**Fig 6. Example of the portrait backings and direct resizing feedback from a healthcare worker (HCW).**

regardless of size (Fig 3B). The barriers to infection control were low, as only 11% "*agree*" or "*strongly agree*" that the portrait badge interferes with PPE donning and doffing and 2% "*disagree*" or "*strongly disagree*" that the portrait badge is easy to clean (Fig 3C). 89.5% of HCWs found that the PPE portraits made a difference in their experiences with patients/residents and 74% found the same for their colleagues or team members (Fig 3D). The pre- and post-effect of the portraits, led to a 37.5% greater likelihood that HCWs felt "*connected*" or "*very connected*" to patients/residents (Fig 3E and 3F).

The quantitative analysis was also done by professions and divided into 2 sets: HCWs (trainees, therapists, nurses, and doctors) that had a sample size of N ≥ 29 (Fig 4A) and HCWs (volunteers, others, admin, and research) with a N ≤ 14 (Fig 4B). Regardless of profession, HCWs "*definitely would*" or "*probably would*" recommend the badge to others (Fig 4A and 4B; minimum to maximum: 91–100%), "*strongly agree*" or "*agree*" that the badge is easy to clean (88–100%), most felt that the badge made more of a difference with patients/residents (85–100%) than for their colleagues or team members (60–100%), and the pre- and post-effect of the portraits increased the percentage of people feeling "*connected*" or "*very connected*" in Fig 4A by 30–51% and Fig 4B by 21–83%, with only 1 of 186 person responding "*difficult to form a connection*" post badge implementation. As for the differences, trainees and doctors reported the lowest portrait usage responding they "*usually*" or "*always*" wore them 53% and 59% of the time (53–92%). Nurses reported the lowest percentage responding "*strongly disagree*" or "*disagree*" to the badge interfering with donning and doffing (42.5%) relative to the rest (Fig 4A not including nurses, 58–73% and Fig 4B 50–67%), though this didn't interfere with their usage (74%). Doctors and volunteers seemed to be outliers in the portraits making a difference in their experience with their colleagues or team members (62% and 60% respectively) relative to others (Fig 4A not including doctors, 70–83% and Fig 4B not including volunteers 75–100%).

Overall for the thematic qualitative analysis (Fig 5), 70% or more of the comments were rated as positive, with less than 5% of comments being rated as negative. Many expressed that the portraits were appreciated particularly by younger and older patients, and were ideal when

dealing with the cognitively impaired, though may present a hazard in certain psychiatric contexts. For their fellow colleagues, responders said the portraits were greatly beneficial due to the constant flux of new masked HCW faces due to staffing increases, HCW turnover, or shifting of personnel from different units and/or institutions. Many respondents expressed being recognized quicker. Even in situations where respondents didn't receive comments, they noticed patients and colleagues looking at their portraits. In some instances, HCWs were made uncomfortable by patients or colleagues as they would receive remarks regarding their appearance that they felt they wouldn't normally get when unmasked.

Furthermore, for the picture resizing that was necessary to have the photo align with our portrait sizing options and to focus on the HCW's facial features (Fig 1), we received informal feedback from a HCW who demonstrated their own resizing preference (Fig 6). This was representative of both informal and formal feedback that indicated that HCWs generally preferred minimal cropping/resizing for submitted photos.

Example of 3 bilingual portrait backings from the 3 most requested regions, with their location specific QR codes, shortened and branded URLs, and supporters. PPE Portraits Canada received direct feedback from an individual regarding their resizing preferences.

## Discussion

PPC serves as an example of how highly accessible online platforms can be incorporated into institutional logistics and workflows to maximize efficiencies of scale, save time, and increase the impact of healthcare initiatives, all while limiting contact and respecting infection control regulations during a pandemic. It also demonstrates how simple medical art interventions can have a measurable and large-scale impact on patients and HCWs, especially given the fear experienced by patients; one study showed that 66% and 18% of patients going to a drive-through COVID-19 testing centre experienced moderate and severe levels of fear respectively [7].

From the perspective of HCWs, PPE portraits did not interfere with infection control (Fig 3C). The vast majority of HCWs felt there was a difference in their experience with patients/residents/colleagues (Fig 3D) and felt more connected with their patients/residents (Fig 3E). HCWs seemed to prefer the smaller portrait size. Though this study could not determine the reasons for this empirically, this is likely due to it not interfering as much with daily tasks and HCW familiarity with the sizing, as the smaller format is similar to standard issue hospital and governmental identification. In settings where there is a low concentration of PPE portraits, healthcare professionals may not want to stand out with a large badge. While there were some differences in responses by profession (Fig 4A and 4B), these need to be investigated with larger sample sizes, with less generalizable groupings, and controlling for profession, setting, and/or individual behaviours (see limitations).

This is the largest study to date on PPE Portraits with 186 participants, compared to the previous of 176 [8]. It is a multi-centered study with feedback from numerous viewpoints: different provinces, cities, institutions, and professions in a variety of settings. The survey sample was representative of the portrait requests and proportional in terms of geography and portrait size. Despite portraits being offered remotely, the diffuse and largely standardized distribution system across Canada allowed for many strong and positive results. We believe these results could have been further improved with additional optimization of the portrait size, picture resizing, and by having more local HCW champions that would increase the feedback and customization of portraits. This would also increase the geographical concentration of portraits, which should address the reports of HCWs' feelings of shyness due to being one of a few to wear a portrait in their setting.

## Limitations

Some limitations of the study include the absence of a standardized timeline for the pilot survey rollout, thus PPE portrait exposure and utilization varied greatly, and our response rate was low. This low response rate necessitated the broad grouping of professions (i.e. trainees, therapists, other). The pre- and post-analysis of the PPE Portraits is subject to recall bias and having a HCW control group, as well as control and non-control patient feedback, would have benefited the study greatly. Measuring the following would have been beneficial in terms of controlling for variables that vary depending on profession, setting, and/or individual behaviours, namely the frequency of: PPE donning/doffing, cleaning their portrait, and interacting with patients/residents and colleagues or team members. Associating a numerical value to the qualitative responses at survey creation and correlating them to these aforementioned measures would have allowed for further insights into optimizing the impact that PPE portraits have in healthcare.

## Practical implications

Future initiatives should consider:

1. A smaller format portrait: 20% more HCWs wore their portraits "*usually*" or "*always*" relative to the larger portrait (Fig 3A) and 14% more HCWs would recommend the portrait (Fig 3B). Future studies may want to consider doing a randomized crossover trial to further examine portrait size preferences and reasoning for this preference.

2. Investigating the following:

   a. HCW and patient preferences regarding inclusion of the HCW's name and/or title on the PPE portrait badge. Though these wouldn't be included directly on the photo, this would limit space on the badge dedicated for the photo itself. In addition, HCWs that have different roles, may need multiple portraits. Ideally, hospital identification should place a greater emphasis on HCWs facial features and include a badge buddy for easy facial, name, and professional identification that can act as a PPE portrait at any time.

   b. The ideal resizing of portraits that balances both the preferences of HCWs and patients. HCWs may be self-conscious wearing a portrait that emphasizes their face too greatly, and thus may be hesitant to wear it. Whereas patients may not easily see a smaller format portrait that does not emphasize facial features. HCWs could be given the option to resize their own portrait and/or have them approve the resizing prior to printing. However, this comes at the expense of HCW and volunteer time. In addition, most of our portraits were single-sided, with a back meant for sponsorship to help fund the portraits and QR codes/links to aid in referral. This means that the headshot itself may not be visible if the portrait swings around and the balance between funding the project, aiding referrals, and having the picture visible regardless of the badge's orientation need to be determined. The most optimal proposal would be to have a double-sided badge, whereby the picture is reduced in size to make room for sponsorship, QR codes, and URLs.

   c. What picture format is preferred by HCWs and patients. Depending on the cultural or situational context, smiling may not be seen as appropriate [4]. Some healthcare institutions requested that their staff not smile in their portrait, to not have patients think their provider is laughing at them. One HCW expressed dissatisfaction with this request and did not use their portrait as a result. In addition, many pictures consisted of non-professional headshots on non-plain backgrounds. While this increased accessibility whereby

any HCW could submit their own picture of their choosing at any time, without the logistical and resource-intensive need of having a photographer take photos, the portrait background may become distracting or be seen as informal. On the other hand, the non-professional aspect of the photos may humanize HCWs to both patients and their colleagues and may be more likely to initiate a conversation relative to the formal nature of hospital identifications. In some cases, HCWs included photos posing with their pets, which may have varying effects on patients and colleagues.

d. The barriers that prevent HCWs from requesting a portrait and how to overcome these. There were numerous comments about colleagues asking where they could get their own portrait, but we do not know the conversion rate on these referrals. In this regard, PPC has never received any formal or informal feedback expressing that the process of obtaining a portrait was difficult. Many HCWs expressed feelings of self-consciousness or not having a recent photograph that they wish to display. Increasing the concentration of portraits within units and hospitals may reduce this hesitancy. Some believed that the portraits should continue post-pandemic and should be integrated into areas of the hospital that regularly use PPE.

## Conclusion

Overall, PPE Portraits should be considered the standard of care whenever PPE is used. Future research should not only optimize the design of portraits, but further assess the impact of PPE Portraits on HCWs and patients alike.

## Supporting information

**S1 Video. Manuscript summary.**
(MP4)

**S1 Table. Response type, themes, and subthemes for the thematic analysis.**
(DOCX)

**S1 File. File: (n = 186) PPC survey.**
(CSV)

## Acknowledgments

We would like to thank (1) all of the study's supporters (2) Dr. Cati G. Brown-Johnson, Natalie Coyle, Amber Linkenheld-Struk, Nadia Radovini, Alex Kiss, and Sara Morassaei for their contribution to the survey development and/or analysis, (3) the PPE Portraits Canada executive team for their volunteer work, (4) all frontline workers who participated and promoted the project and in particular Caroline Coutu, and (5) Mary Beth Heffernan and David Gutnick who shared their resources and experiences regarding the PPE portrait project.

## Author Contributions

**Conceptualization:** Ana Seara, Adamo A. Donovan.

**Data curation:** Laura Rendon, Tarek Taifour, Cynthia R. Ventrella, Adamo A. Donovan.

**Formal analysis:** Laura Rendon, Tarek Taifour, Cynthia R. Ventrella, Adamo A. Donovan.

**Funding acquisition:** Cynthia R. Ventrella, Ana Seara, Adamo A. Donovan.

**Methodology:** Ana Seara, Adamo A. Donovan.

**Project administration:** Laura Rendon, Cynthia R. Ventrella, Ana Seara, Adamo A. Donovan.

**Supervision:** Ana Seara, Adamo A. Donovan.

**Visualization:** Laura Rendon, Tarek Taifour.

**Writing – original draft:** Adamo A. Donovan.

**Writing – review & editing:** Laura Rendon, Tarek Taifour, Cynthia R. Ventrella, Ana Seara, Adamo A. Donovan.

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
