## [Decision Letter · Decision Letter 0]

17 Nov 2023

PONE-D-23-26716Personal Protective Equipment Portraits Canada (PPC) – Humanization and surveying mask-wearing nationallyPLOS ONE

Dear Dr. Donovan,

Thank you for submitting your manuscript to PLOS ONE. After careful consideration, we feel that it has merit but does not fully meet PLOS ONE’s publication criteria as it currently stands. Therefore, we invite you to submit a revised version of the manuscript that addresses the points raised during the review process.

The paper presents interesting and relevant results of the impacts suffered by healthcare professionals and patients due to the Covid-19 Pandemic. However, some corrections are necessary before it can be considered for publication according to the opinion of the reviewers and the academic editor:

1) Include information that answers the following questions:

a) Was it possible to establish any relationship between the feedback obtained in the research and the profession of the participants?

b) Was there any relationship between this factor and the experience with patients and colleagues?

2) Include in the paper a summary of the research graphics, also listing the professions of the participants.

3) Present the results of the thematic qualitative analysis in graphic form, highlighting the percentage of positive, negative and indifferent responses, and for each of them, the respective percentage of responses according to the themes and subthemes presented in Supplementary Table 2.

4) Present the statistical analysis of the data obtained to support the discussion.

5) Add the figures and tables to the body of the work, and discuss the data presented further, as they are still a little confusing.

Please submit your revised manuscript by Jan 01 2024 11:59PM

. If you will need more time than this to complete your revisions, please reply to this message or contact the journal office at plosone@plos.org. Please include the following items when submitting your revised manuscript:A rebuttal letter that responds to each point raised by the academic editor and reviewer(s). You should upload this letter as a separate file labeled 'Response to Reviewers'.A marked-up copy of your manuscript that highlights changes made to the original version. You should upload this as a separate file labeled 'Revised Manuscript with Track Changes'.An unmarked version of your revised paper without tracked changes. You should upload this as a separate file labeled 'Manuscript'.If applicable, we recommend that you deposit your laboratory protocols in protocols.io to enhance the reproducibility of your results. Protocols.io assigns your protocol its own identifier (DOI) so that it can be cited independently in the future. For instructions see: https://journals.plos.org/plosone/s/submission-guidelines#loc-laboratory-protocols. Additionally, PLOS ONE offers an option for publishing peer-reviewed Lab Protocol articles, which describe protocols hosted on protocols.io. Read more information on sharing protocols at https://plos.org/protocols?utm_medium=editorial-email&utm_source=authorletters&utm_campaign=protocols.

We look forward to receiving your revised manuscript.

Kind regards,

Angelo Marcelo Tusset

Academic Editor

PLOS ONE

Journal Requirements:

"The authors received a Practiced-based Research and Innovation (PBRI) Special Seed Grant from Sunnybrook Health Sciences Centre to cover the costs of the survey implementation and Survey Monkey subscription.

To pay for all fees associated with PPE Portraits Canada’s informational technology backend and the logistics supporting the request, purchase, and delivery of portraits, the authors received the following:

- 2020 McGill Mary H Brown Fund 

- 2020 TakingITGlobal #RisingYouth grant

- 2020 Surrey Memorial Hospital Engagement Project Fund

- 2021 University of Toronto Pillar Sponsorship Program

- 2021 Research Institute of the McGill University Health Centre (MUHC) Desjardins Centre for Advanced Training (DCAT) Trainee-led Initiative Funding

- 2021 Ontario Medical Student Association (OMSA) Innovator Grant "

Please include this amended Role of Funder statement in your cover letter; we will change the online submission form on your behalf

"I have read the journal's policy and the authors of this manuscript have the following competing interests: This paper mentions several softwares and programs utilized by the authors and the PPE Portraits Canada organizing team. Both parties have benefitted from non-profit or COVID-19 response discounts and/or free premium subscriptions to use these programs at scale. However, these are non-exclusive and none of the authors have received any monetary compensation nor are they affiliated in any other way with the companies behind these products. Exceptionally, PPC received an exclusive sponsorship from a local printing sponsor in Montréal (The Business Box) that is mentioned in the article to receive PPE Portraits at cost. The authors are in no way obligated to mention these programs but are doing so to inspire others and facilitate the scaling of other initiatives."

7. In your Data Availability statement, you have not specified where the minimal data set underlying the results described in your manuscript can be found. PLOS defines a study's minimal data set as the underlying data used to reach the conclusions drawn in the manuscript and any additional data required to replicate the reported study findings in their entirety. All PLOS journals require that the minimal data set be made fully available. For more information about our data policy, please see http://journals.plos.org/plosone/s/data-availability.

8. Your ethics statement should only appear in the Methods section of your manuscript. If your ethics statement is written in any section besides the Methods, please delete it from any other section. 

9. We note that Figures 1 and Supp Video1 in your submission contain copyrighted images. All PLOS content is published under the Creative Commons Attribution License (CC BY 4.0), which means that the manuscript, images, and Supporting Information files will be freely available online, and any third party is permitted to access, download, copy, distribute, and use these materials in any way, even commercially, with proper attribution. For more information, see our copyright guidelines: http://journals.plos.org/plosone/s/licenses-and-copyright.

a. You may seek permission from the original copyright holder of Figures 1 and Supp Video 1 to publish the content specifically under the CC BY 4.0 license. 

10. Please include captions for your Supporting Information files at the end of your manuscript, and update any in-text citations to match accordingly. Please see our Supporting Information guidelines for more information: http://journals.plos.org/plosone/s/supporting-information. 

11. We note that Figure 4 includes an image of a participant . 

Additional Editor Comments:

The paper presents interesting and relevant results of the impacts suffered by healthcare professionals and patients due to the Covid-19 Pandemic. However, some corrections are necessary before it can be considered for publication according to the opinion of the reviewers and the academic editor:

1) Include information that answers the following questions:

a) Was it possible to establish any relationship between the feedback obtained in the research and the profession of the participants?

b) Was there any relationship between this factor and the experience with patients and colleagues?

2) Include in the paper a summary of the research graphics, also listing the professions of the participants.

3) Present the results of the thematic qualitative analysis in graphic form, highlighting the percentage of positive, negative and indifferent responses, and for each of them, the respective percentage of responses according to the themes and subthemes presented in Supplementary Table 2.

4) Present the statistical analysis of the data obtained to support the discussion.

5) Add the figures and tables to the body of the work, and discuss the data presented further, as they are still a little confusing.

Reviewers' comments:

Reviewer's Responses to Questions

**Comments to the Author**

1. Is the manuscript technically sound, and do the data support the conclusions?

Reviewer #1: Partly

Reviewer #2: Yes

2. Has the statistical analysis been performed appropriately and rigorously? 

Reviewer #1: No

Reviewer #2: Yes

3. Have the authors made all data underlying the findings in their manuscript fully available?

Reviewer #1: Yes

Reviewer #2: Yes

4. Is the manuscript presented in an intelligible fashion and written in standard English?

Reviewer #1: Yes

Reviewer #2: Yes

5. Review Comments to the Author

Reviewer #1: The work entitled “Personal Protective Equipment Portraits Canada (PPC) – Humanization and surveying mask-wearing nationally” brings an interesting and relevant contribution, given the impacts suffered by healthcare workers and patients due to the Covid-19 Pandemic. The proposal is relevant, the results are interesting and the work is well organized.

Before publication, a more detailed analysis of the data obtained must be carried out.

1) Was it possible to establish any relationship between the feedback obtained in the survey and the profession of the participants? Was there any relationship between this factor and the experience with patients and colleagues? Present the summary of survey graphs also listing the professions of the participants.

2) Also present the results of the thematic qualitative analysis in graphic form, showing the percentage of positive, negative and indifferent responses, and for each of them, the respective percentage of responses according to the themes and subthemes presented in the Supplemental Table 2.

3) Present the statistical analysis of the data obtained to support the discussion.

Reviewer #2: The article has separate figures and data. I think the author should put the figures together with the text and the data should be better commented, as they are still a bit confusing. If the data is in the text, readers can see it and interpret it more easily.

6. PLOS authors have the option to publish the peer review history of their article (what does this mean?). If published, this will include your full peer review and any attached files.

Reviewer #1: No

Reviewer #2: No

---

## [Author Response · Author response to Decision Letter 0]

8 Jan 2024

Comments

1) Include information that answers the following questions: 

a. Was it possible to establish any relationship between the feedback obtained in the research and the profession of the participants? 

b. Was there any relationship between this factor and the experience with patients and colleagues? 

2) Include in the paper a summary of the research graphics, also listing the professions of the participants.

This is a really interesting idea. We’ve investigated the relationship between professions and the feedback/experience with patients and colleagues and summarized these with the inclusion of Figure 4.

3) Present the results of the thematic qualitative analysis in graphic form, highlighting the percentage of positive, negative and indifferent responses, and for each of them, the respective percentage of responses according to the themes and subthemes presented in Supplementary Table 2. 

Thank you for this suggestion as we agree presenting the thematic qualitative analysis in graphic form will be easier for readers to interpret.

We’ve done this and is presented in Figure 5.

4) Present the statistical analysis of the data obtained to support the discussion. 

We are not sure what the reviewers mean by this as the survey data is qualitative in nature and we don’t believe a statistical analysis to determine if the PPE portraits made a significant difference can be done. We did add in our limitations “associating a numerical value to the qualitative responses at survey creation” would have allowed for such analyses.

Do the reviews have a statistical analysis in mind that they would like us to run on the data? Any recommendations would be much appreciated.

5) Add the figures and tables to the body of the work, and discuss the data presented further, as they are still a little confusing.

We have added the figures and tables to the text and have added discussion points to clarify and deepen the results presented.

---

## [Decision Letter · Decision Letter 1]

17 Jan 2024

Personal Protective Equipment Portraits Canada (PPC) – Humanization and surveying mask-wearing nationally

PONE-D-23-26716R1

Dear Mr. Donovan,

We’re pleased to inform you that your manuscript has been judged scientifically suitable for publication and will be formally accepted for publication once it meets all outstanding technical requirements.

Kind regards,

Angelo Marcelo Tusset

Academic Editor

PLOS ONE

---

## [Editor Report · Acceptance letter]

14 Feb 2024

PONE-D-23-26716R1 

PLOS ONE

Dear Dr. Donovan, 

I'm pleased to inform you that your manuscript has been deemed suitable for publication in PLOS ONE. Congratulations! Your manuscript is now being handed over to our production team.

Kind regards, 

on behalf of

Professor Angelo Marcelo Tusset 

Academic Editor

PLOS ONE